# Attitudes towards Vaccines, Intent to Vaccinate and the Relationship with COVID-19 Vaccination Rates in Individuals with Schizophrenia

**DOI:** 10.3390/vaccines10081228

**Published:** 2022-07-31

**Authors:** Stéphane Raffard, Sophie Bayard, Margot Eisenblaetter, Jérôme Attal, Christelle Andrieu, Isabelle Chereau, Guillaume Fond, Sylvain Leignier, Jasmina Mallet, Philippe Tattard, Mathieu Urbach, David Misdrahi, Yasmine Laraki, Delphine Capdevielle

**Affiliations:** 1Laboratory EPSYLON, Paul Valéry University Montpellier 3, CEDEX 5, 34199 Montpellier, France; sophie.bayard@univ-montp3.fr (S.B.); margot.eisenblaetter@gmail.com (M.E.); y-laraki@chu-montpellier.fr (Y.L.); 2University Department of Adult Psychiatry, CHU Montpellier, 34000 Montpellier, France; j-attal@chu-montpellier.fr (J.A.); p-tatard@chu-montpellier.fr (P.T.); d-capdevielle@chu-montpellier.fr (D.C.); 3Pôle Psychiatrie, Centre Expert Dépression Résistante et Schizophrénie Fonda Mental, CHU La Conception, 13005 Marseille, France; christelle.andrieu@ap-hm.fr (C.A.); guillaume.fond@gmail.com (G.F.); 4CEReSS-health Service Research and Quality of Life Center, Aix-Marseille University, 27 Boulevard Jean-Moulin, 13005 Marseille, France; 5CHU Clermont-Ferrand, Service de Psychiatrie B, Université Clermont Auvergne, 63000 Clermont-Ferrand, France; ichereau@chu-clermontferrand.fr; 6Centre Référent de Réhabilitation Psychosociale et de Remeédiation Cognitive (C3R), CH Alpes Iseère, 38400 Saint-Martin-d’Hères, France; sleignier@ch-alpes-isere.fr; 7AP-HP, Department of Psychiatry, Louis Mourier Hospital, 92700 Colombes, France; jasmina.mallet@aphp.fr; 8INSERM UMR1266, Institute of Psychiatry and Neuroscience of Paris, University Paris Descartes, 75006 Paris, France; 9Faculté de Médecine, Université Paris Diderot, Sorbonne Paris Cité, 75013 Paris, France; 10Department of Adult Psychiatry and Addictology, Versailles Hospital, Centre Hospitalier de Versailles, 177 rue de Versailles, 78157 Le Chesnay, France; murbach@ch-versailles.fr; 11DisAP-DevPsy-CESP, INSERM UMR1018, University of Paris-Saclay, University of Versailles Saint-Quentin-En-Yvelines, 94807 Villejuif, France; 12Department of Adult Psychiatry, Charles Perrens Hospital, University of Bordeaux, Laboratory of Nutrition and Integrative Neurobiology (UMR INRA 1286), 33000 Bordeau, France; dmisdrahi@ch-perrens.fr; 13IGF, University Montpellier, CNRS, INSERM, 34000 Montpellier, France

**Keywords:** COVID-19, attitudes, vaccination, schizophrenia

## Abstract

Schizophrenia patients are at high risk of developing severe COVID-19 outcomes but recent evidence suggests that they are under-vaccinated. This study explored the role of potential attitudinal barriers by comparing schizophrenia patients with participants from the general population regarding COVID-19 vaccination rates, general attitudes towards vaccines, and willingness to take a COVID-19 vaccine. We conducted a cross-sectional study between April 2021 and October 2021. A total of 100 people with schizophrenia and 72 nonclinical controls were recruited. In our study, individuals with schizophrenia were under-vaccinated, despite similar general attitudes towards vaccination and higher willingness to be vaccinated against COVID-19 compared to nonclinical participants. In patients, negative attitudes toward vaccines were related to higher levels of negative psychotic symptoms and higher levels of paranoid ideation. As a whole, participants with more negative attitudes towards vaccines were less likely to be vaccinated against COVID-19 and had lower levels of trust in institutions. Vaccine hesitancy does not appear to be a major barrier for COVID-19 vaccine uptake amongst people with schizophrenia. This study suggests that disparities in COVID-19 vaccination rates in schizophrenia do not seem related to attitudinal but rather structural barriers.

## 1. Introduction

The worldwide pandemic caused by the COVID-19 (SARS-CoV-2) has strongly impacted individuals’ lives with dramatic economical and global health consequences. More than 250 million cases and 5 million deaths worldwide were reported as of 30 November 2021 (WHO). An accumulating body of research pointed out that individuals with severe mental illnesses, including schizophrenia spectrum disorders, mood, and substance use disorders are at higher risk of developing serious COVID-19 outcomes, including hospitalizations and death [1,2,3,4]. Among the different public health measures that have been implemented to curb this health crisis, the recent development of effective vaccines constitutes a decisive step in the successful control of this pandemic. As such, promoting vaccination access in people with schizophrenia is a public health priority [5]. However, and despite the demonstrated effectiveness of vaccines for reducing the mortality and morbidity of COVID-19 in this population, a cross-sectional and a longitudinal study reported an association between this pathology and lower vaccination rates [6,7]. These studies are in line with what has already been observed in other preventive vaccines, such as the influenza vaccine [8,9].

Based on past global experiences from infectious disease control, researchers have identified two types of barriers for vaccination: structural and attitudinal [10]. Structural barriers correspond to “systemic issues that may limit the ability of individual persons to access a vaccine service” [10]. Regarding schizophrenia, several structural barriers have been identified, including cost, limited access to online health information, awareness of services, or absence of medical recommendation [5,9].

The second type of barriers for vaccination refers to attitudes, which can be defined as a set of emotions, beliefs, and behaviors toward a particular object, person, thing, or event [11,12]. Similar to all existing vaccines, it has been shown that the most important attitudinal barriers to receiving a COVID-19 vaccine are a general mistrust in the benefits and safety of vaccines, as well as concerns about their unforeseen secondary effects [13]. As a result, vaccination attitudes are considered as multidimensional rather than a unidimensional construct [10,14]. Recently and within the context of the COVID-19 pandemic, two large-scale studies have shown that both conspiracy beliefs and persecutory ideas, two psychological features that are highly present in individuals with schizophrenia, were positively associated with COVID-19 vaccine hesitancy and resistance in the general population [15,16]. Finally, our research team recently showed that impaired competence to consent to a COVID-19 vaccine was associated with lower vaccination rates in people with schizophrenia [17].

However, the majority of the literature on attitudes towards vaccination focuses on the general population and very few studies have explored these attitudes in the context of severe mental disorders. The first and only study that has assessed attitudes towards vaccination (excluding the COVID-19 pandemic) showed that individuals with schizophrenia perceived vaccination against influenza as effective as non-clinical controls, but judged more risky for adverse reactions [18]. Regarding COVID-19 vaccination, two recent studies in China and Israel showed that individuals with various mental disorders had similar positive attitudes for the COVID-19 vaccine compared to the general population [19,20]. However, the majority of their sample was composed of people diagnosed with anxiety disorders (including generalized anxiety disorder and panic disorder) and depression (major depressive disorder) with only a small sample (*N* = 16) of individuals diagnosed with psychotic disorders [19,20]. In contrast, a third study showed that individuals with schizophrenia had more negative attitudes than the general population towards COVID-19 vaccination [21]. However, in their study, attitudes towards vaccination was measured using a single question “Do you intend to be vaccinated against COVID-19 in the future?”, which, rather than assessing attitudes, this question measures the intention to be vaccinated [21]. As a result, the attitudes of people with schizophrenia spectrum disorders regarding vaccination remains largely unexplored. Moreover, understanding how attitudes are related to both intent to be vaccinated and vaccination rates has not yet been investigated in people with schizophrenia despite evidence that COVID-19 vaccination is crucial for this population [5,22].

Consequently, the aims of this study were: (1) to assess general attitudes towards vaccines using a multidimensional scale in individuals with schizophrenia compared to a nonclinical control group, (2) to determine vaccination rates as a function of general attitudes towards vaccines in individuals with schizophrenia, and (3) to investigate the clinical, cognitive and psychological determinants of attitudes toward vaccines in general and towards COVID-19 vaccination in individuals with schizophrenia.

## 2. Materials and Methods

### 2.1. Sample

One-hundred participants with a diagnosis of schizophrenia were recruited following the same protocol in four independent sites in France between April 2021 to October 2021 (CHU Montpellier, CHU Grenoble, AP-HP Marseille, CHU Versailles). Inclusion criteria for patients were: (a) ages between 18 and 65 years, (b) a DSM-5 diagnosis of schizophrenia, and (c) adequate proficiency in French. Exclusion criteria for all participants were: (a) known neurological disease, (b) history of learning disability/developmental disorder, or (c) substance abuse in the past month (other than cannabis or tobacco). For the nonclinical participants, there was an additional exclusion criterion: no personal lifetime history diagnosis of a psychotic or affective disorder. To ensure the absence of psychotic disorders in the non-clinical participants, we used the 7th version of the DSM-5 Mini-International Neuropsychiatric Interview. Non-clinical participants who had a family member with either bipolar or schizophrenia disorder were also excluded. In total seventy-two controls were recruited from the general population in Montpellier. The demographic data for all participants are shown in Table 1.

### 2.2. Assessments

*Clinical and cognitive assessments.* The positive and negative syndrome scale (PANSS) is a 30-item scale designed to assess specific symptomatology in psychotic disorders in individuals diagnosed with a schizophrenia disorder [23]. Each item is based on a 1 to 7 rating-scale and overall scores include a positive, negative, general psychopathology, and total score. Paranoid ideations were assessed in both groups using the short version of the Green Paranoid Thoughts Scale (GPTS), which has recently been validated in French [24]. This version of the GTPS comprises of 8 items based on a 5-point rating scale ranging from 1 (not at all) to 5 (totally). Global cognitive functioning was assessed in both groups using the Montreal Cognitive Assessment (MoCA) [25].

*Vaccination status and health-related indicators.* COVID-19 vaccination status was recorded for all participants as either no vaccination or first dose of vaccine/fully vaccinated (i.e., three doses). Participants were also asked if they or any of their immediate family members had any of the following underlying health conditions: diabetes, lung disease, or cardiovascular conditions. We also recorded whether the participants had a confirmed or suspected case of COVID-19 infection.

*Attitudes toward vaccines*. Negative general attitudes towards vaccines were measured using the 12-item Vaccination Attitudes Examination Scale (VAX, 14). All participants were asked to focus on vaccines in general rather than specifically on a COVID-19 vaccine. Responses on the VAX are rated on a six-point scale ranging from 1 (strongly agree) to 6 (strongly disagree). Four subscale-scores were calculated: (1) mistrust of vaccine benefit (e.g., “I can count on vaccines to remove serious infectious diseases”, reverse item), (2) worries about unforeseen future effects (e.g., “I am concerned about the unknown long-term effects of vaccines”), (3) concerns about commercial profiteering (e.g., “The authorities encourage vaccination for financial reasons, not for the health of the people”), and (4) preference for natural immunity (e.g., “Natural immunity lasts longer than than that acquired by vaccination”). Note that higher scores indicate higher negative attitudes towards vaccines. The French version of the VAX showed good internal consistency, construct validity and test–retest reliability [26].

*Intent to take a COVID-19 vaccine.* For the unvaccinated participants, we also measured the intent to take a COVID-19 vaccine. We used a single item question taken from the IPSOS-WEF COVID-19 VACCINE GLOBAL SURVEY JANUARY 2021—GLOBAL PR: “If a vaccine for COVID-19 is available for me, I would get it”. Participants were asked to respond to this item on a 4 point-scale ranging from 1 (strongly disagree) to 4 (strongly agree).

*Trust in institutions*. In order to evaluate the trust in institutions, we used the same methodology as Murphy et al. [16]. Participants were asked to indicate the level of trust they had in political parties, the parliament, the government, the police, the legal system, scientists, doctors, and other health professionals. This questionnaire is composed of 7 items and responses were scored on a five-point Likert scale ranging from “do not trust at all” (1) to “completely trust” (5). A total score was computed whereby higher scores indicated more trust in institutions.

### 2.3. Procedure

For the clinical participants, nursing staffs of the different sites identified all incoming outpatients with a diagnosis of schizophrenia and were approached for participation to the study. In addition, the nursing staff regularly checked the electronic medical records and consultation boards for any potential participants. A total of 122 individuals with schizophrenia were screened for eligibility during the study period, amongst which 120 eligible patients were approached for an interview and a full description of the study was given. Mental capacity to participate in this study was based on the clinical assessment and a thorough review of the patients’ clinical notes. All participants provided informed and written consent. Additional information was collected from the participants and they were then administered the measures for a total duration of approximately one hour. Eighteen patients declined participation, and twelve did not complete the entire protocol. Therefore, our final clinical sample included 100 individuals with a diagnosis of schizophrenia. For the nonclinical group, participants were recruited from the general population using posters on the hospitals’ noticeboards and through word of mouth. This study was conducted in accordance to the ethical standards described by the Medical Research Involving Human Subjects Act (WMO) and was approved by the hospital’s institutional review board (IRB ID: 202100768).

### 2.4. Statistical Analysis

Statistical analyses were performed with the Jamovi statistical computer software [The jamovi project (2021). jamovi. (Version 1.6) Retrieved from http://www.jamovi.org, accessed on 27 July 2022]. Means and standard deviations were computed for continuous variables and categorical variables were expressed in percentages. Absolute values for skewness and kurtosis greater than 3 and 20, respectively, were used to test normality [27]. The winsorizing method was used to process any outlier scores on questionnaires. An outlier score was considered when its z-score had an absolute value of |3.29|, which is 0.01% of the distribution. This method consists of replacing outlier scores by assigning a value on a unit smaller or larger than the next most extreme (non-outlier) score in the distribution [28].

The internal consistency of the French version of the VAX was examined with the corrected item-total correlation and Cronbach’s alpha (good internal consistency was considered 0.7 > α < 0.9). Group differences in demographic variables, clinical variables, and scores on the questionnaires were analyzed using independent-sample Student’s t-test and analysis of variance for parametric variables. Chi-square tests were used for categorical variables. In addition, we calculated the eta squared η^2^, the Cohen’s d′ and the Phi-coefficient as measures of effect size. The effect size was considered small (η^2^ = 0.01; d = 0.2; Phi = 0.10), medium (η^2^ = 0.06; d = 0.5; Phi = 0.30), or large (η^2^ = 0.14; d = 0.8; Phi = 0.5). To test for the potential relationships between variables in the whole sample and per group, we used Pearson correlations with rs of 0.10, 0.30, and 0.50 defined as small, medium, and large effect size, respectively. All analyses were conducted with a significance threshold of α ≤ 0.05, two-tailed.

## 3. Results

### 3.1. Preliminary Analyses

All data from questionnaires had satisfactory skewness (0.01 to 1.38) and kurtosis (−0.92 to 3.24) values, suggesting a normal distribution. In the total sample and for all items of our variables of interest (n = 1548), only 10 values were identified as univariate outliers (10/1548) * 100 = 0.64%. Except for the worries about unforeseen future effects (α = 0.55) subscale, all Cronbach’s alpha for the other subscales of the VAX were considered good: mistrust of vaccine benefit (α = 0.86), concerns about commercial profiteering (α = 0.74), and preference for natural immunity (α = 0.75) subscales. The item–total correlations for the 14 items of the VAX ranged from 0.35 to 0.66, with a mean of 0.54.

### 3.2. Demographic Variables

As documented in Table 1, patients were older than controls (*p* < 0.001, d = −0.63) and overall had a lower level of education (*p* < 0.001, d = 0.56). No significant group difference was noted in terms of gender (*p* = 0.15).

### 3.3. Clinical and Cognitive Assessments

All patients were under antipsychotic treatment. Based on the GPTS results, patients reported higher level of paranoid ideations compared to the controls (*p* = 0.02, d = −0.33). The clinical group also showed lower global cognitive functioning on the MOCA, compared to the non-clinical group (*p* < 0.001, d = 0.93) (Table 1).

### 3.4. Vaccination Status, Health-Related Indicators, and Intention to Vaccinate

A statistical trend was noted with respect to the proportion of vaccinated participants, showing lower rates of vaccination in patients, compared to controls (respectively, 64% versus 77.8%, *p* = 0.07, Phi = 0.15). Note that neither age, education, nor global cognitive functioning was associated with vaccination status in either group (all Ps > 0.20). The proportion of participants infected or suspected of having been infected with COVID-19 was not significantly different in the two groups (*p* = 0.15). Patients reported significantly higher levels of underlying health conditions compared to controls (*p* = 0.01). In both groups, no association was found between vaccination status and health-related indicators (all Ps > 0.40). Finally, among the non-vaccinated participants, the proportion of individuals intending to be vaccinated was significantly higher in patients compared to controls (*p* = 0.03, Phi = 0.29).

### 3.5. Vaccination Status, Demographic Variables, Health-Related Indicators, and Clinical and Cognitive Assessments

In both patients and in controls, no association was found between vaccination status, demographic variables (i.e., age, education, and gender) and health-related indicators (all Ps < 0.13). In patients, vaccination status was not significantly associated with global cognitive functioning (*p* = 0.34), psychotic symptoms (all Ps > 0.56) nor paranoid ideations (*p* = 0.13).

### 3.6. Vaccination Status, Attitudes towards Vaccines, Trust in Institutions, and COVID-19 Related Questionnaires

Figure 1 illustrates that there was a significant difference in all VAX dimensions based on vaccination status (all Ps < 0.01). Non-vaccinated individuals had higher negative attitudes towards vaccines across all VAX dimensions. Furthermore, there was a significant group effect (*p* = 0.004, η^2^ = 0.04) for the VAX mistrust of vaccine benefit dimension (Figure 1A). Patients reported higher confidence in the benefits of vaccination compared to controls. No group effect was observed for the other three dimensions of the VAX questionnaire (Figure 1B–D; all Ps > 0.22). Interactions were not significant either (Figure 1; all Ps > 0.10).

Finally, a significant association was noted between vaccination status and trust in institutions questionnaire. Non-vaccinated individuals had lower levels of trust in institutions than did the vaccinated participants (F = 5.7, *p* = 0.003, η^2^ = 0.05, respectively, 21.1 ± 6.7 versus 22.7 ± 5.4). Patients had a significantly higher level of confidence in the institutions than the controls (F = 9.2, *p* = 0.02, η^2^ = 0.03, respectively, 23.1 ± 4.5 versus 20.9 ± 6.5). The interaction between vaccination status and preference for natural immunity group was not significant (F = 1.7, *p* = 0.18).

### 3.7. Associations between Attitudes towards Vaccines, COVID-19 Related Questionnaires, Demographic Variables, Health-Related Indicators, and Clinical and Cognitive Assessments

In both patients and controls, no association was found between attitudes toward vaccines, demographic variables, and health-related indicators (all Ps > 0.10). In patients, the preference for natural immunity VAX dimension was related to higher levels of negative psychotic symptoms (r = 0.31, *p* = 0.01) and to higher levels of paranoid ideation (r = 0.22, *p* = 0.02). Higher levels of paranoid ideation was also associated with the concerns about commercial profiteering VAX domain (r = 0.21, *p* = 0.04).

In both patients and controls, both the VAX mistrust of vaccine benefit (respectively, r =−0.39, *p* < 0.001 versus r =−0.32, *p* = 0.005) and concerns about commercial profiteering VAX dimensions were related to a lower level of trust in institutions (r =−0.23, *p* = 0.02 versus r =−0.51, *p* < 0.001). Finally, only in the controls group was higher level of worries about unforeseen future effects VAX dimension associated to lower levels of trust in institutions (r =−0.32, *p* = 0.005).

## 4. Discussion

Currently, there is evidence that individuals with mental disorders, and particularly individuals with schizophrenia, should be prioritized for COVID-19 vaccination as hospitalization and mortality rates are higher in this population in comparison to the general population [2,7]. Attitudinal barriers towards vaccination might reduce vaccine uptake in this population but these remain largely unexplored, specifically within the context of the COVID-19 pandemic [22,29].

First, general attitudes towards vaccination as measured by the total score of the VAX scale were similar between patients and controls. Patients and controls did not differ regarding beliefs about unforeseen future effects, concerns about commercial profiteering, and preference for natural immunity. However, patients scored lower on the mistrust of vaccine benefit dimension of the VAX compared to the controls. This indicates that our sample of patients (even though they were under-vaccinated) had more positive attitudes regarding vaccination and had a stronger belief that we can rely on vaccines to stop and control the spread of serious infectious diseases than the nonclinical participants. On average, patients felt safer after being vaccinated, considered vaccines to be more effective in stopping serious infectious diseases and felt more protected after getting vaccinated compared to controls. Our results are partially in accordance with Maguire et al. who found that schizophrenia patients considered the vaccine for influenza as being as effective as the control group, but they were also more likely to perceive vaccination as being risky for an adverse reaction [18]. One explanation could be that contrarily to the influenza vaccines, the government and the media throughout the world have heavily promoted COVID-19 vaccination. Due to the severity of this pandemic, the development of vaccines and vaccination campaigns progressed rapidly worldwide, potentially contributing to an overall increase in vaccine confidence. In addition, our results indicate that individuals with schizophrenia (and individuals with other mental disorders) do not necessarily have more negative attitudes towards vaccines than the general population [13,30,31]. This differs from what has been shown in other COVID-19 at-risk populations (e.g., ethnic minority groups and individuals with lower incomes or education) [30,32,33]. One can also hypothesize that the high rate of positive attitudes towards vaccination may partly be explained by the prioritization for vaccination of patients with mental disorders following the publication of high mortality rates of these populations in France [2].

Second, a trend was observed in terms of the proportion of vaccinated participants, which was lower in patients in comparison to the non-clinical controls (respectively, 64% versus 77.8%, *p* = 0.07). Though COVID-19 vaccination rates in people diagnosed with severe mental disorders are to date largely unknown in most countries, a recent study in Israël, which included a large sample of patients, indicated that people with schizophrenia are under-vaccinated for COVID-19 compared to the general population [7]. Very importantly, among non-vaccinated participants, a significantly higher proportion of patients had the intention to be vaccinated compared to nonclinical controls. In other words, despite similar general positive attitudes towards vaccines and higher willingness to be vaccinated (for the non-vaccinated individuals), the rates of vaccination for COVID-19 were lower in our patient group.

Two types of barrier to vaccination have been described in the literature: structural (e.g., poor internet access, shorter life expectancy, geographical distance from vaccination sites, living in more deprived areas) and attitudinal (i.e., beliefs or perceptions that impact the willingness to be vaccinated) [7,10,22,34]. Overall, our results suggest that structural barriers rather than individual barriers might explain the lower vaccination rates in our group of individuals with schizophrenia. This is in line with the fact that although people with schizophrenia are at higher risk than other populations in developing some physical diseases, they are rarely considered as a priority group to receive a vaccination. However, research has shown that when this population is correctly informed, the willingness to be vaccinated is high (see similar results regarding pneumonia and influenza) [8,35].

Third, we found that in both groups, non-vaccinated individuals had higher general negative attitudes towards vaccines across all VAX dimensions. Even though a causal inference between general attitudes towards vaccination and vaccination rates cannot be inferred, this is the first time that such an association is found in schizophrenia. Additionally, this suggests that attitudes toward COVID-19 vaccination might influence vaccination coverage rates, which is the case for other types of vaccines, such as influenza vaccines [36]. Further studies using a longitudinal design are needed to draw definitive conclusions.

Fourth, correlational analyses found negative relationships between positive attitudes towards vaccination and negative symptoms and paranoid ideations, measured using the PANSS and the GPTS, respectively. The symptomatology of individuals with schizophrenia should thus be considered when assessing confidence and intent towards vaccination. The association between paranoid ideation and general negative attitudes towards vaccination in schizophrenia is in accordance with previous studies conducted in the general population, whereby similar associations between conspiracy beliefs, persecutory beliefs, and higher resistance to a COVID-19 vaccine was found [15,16]. However, the fact that general attitudes towards vaccination were similar between patients and controls suggests that individuals with schizophrenia are not additionally impacted by misinformation and anti-vaccination influences, compared to the general population. The association between negative symptoms and vaccination hesitancy appears however difficult to explain and remains elusive. Nevertheless, the desire for restrictions to ease and life to return to normal (e.g., go to the cinema, travel) has been found as the most common reasons for positive attitudes towards COVID-19 vaccine [37,38]. It is likely that the desire to perform such activities do not constitute a strong enough factor in reducing vaccine hesitancy in people with schizophrenia with high levels of negative symptoms. For example, studies have recurrently shown that negative symptoms, and specifically apathy–anhedonia, lead to a reduced capacity in experiencing and anticipating pleasure from activities [39]. A reduction in interpersonal interactions and goal-directed behaviors have also been noted in this population. This could also explain the association found between negative symptoms and vaccination hesitancy.

Finally, we also found that individuals with schizophrenia had significantly higher levels of confidence in the institutions in comparison to the controls. Our results also indicate that non-vaccinated participants (in both groups) had lower levels of trust in institutions, compared to the vaccinated participants. This result is in accordance with recent research that emphasized the role of trust in science as a determinant factor in both vaccine hesitancy and intention to be vaccinated [40,41,42]. Importantly, this result shows similar associations in individuals with schizophrenia.

The current study has several limitations that restrict the interpretation and generalizability of the results. The main limitation is the cross-sectional design of our study. Even though we found an association between negative attitudes and lower vaccination rates in both groups, we cannot conclude a causal relationship between these two variables. Future studies with a longitudinal design are needed. Second, our sample group was composed exclusively of French individuals, limiting the generalizability of our results. There is evidence that in Europe, and particularly in France, trust towards vaccination remains low compared to other continents [38]. The replication of the present study in other countries would allow for a better understanding of attitudes towards vaccination in different populations (including in individuals with mental illness). Finally, we did not take into account other non-structural determining factors that could have contributed to the differences in vaccination rates in our two populations. For example, several studies have explored the risk of developing neuropsychiatric side-effects of vaccination and the potential interactions between vaccines and psychotropic medications [43]. Such side-effects may have contributed to lower vaccine uptake in our clinical sample and may represent an additional limitation to our study.

## 5. Conclusions

The present study found that individuals with schizophrenia have similar general attitudes towards vaccines as the general population. The non-vaccinated patients also endorsed a higher willingness to be vaccinated against COVID-19 compared to the non-vaccinated controls. However, and as a whole, people with schizophrenia were under-vaccinated in our sample suggesting the presence of structural barriers rather than attitudinal barriers in getting vaccinated within this clinical population. Developing targeted vaccination programs is urgently needed for this population [44,45]. Importantly, as the large majority of schizophrenia patients were recruited in the area of Montpellier (south of France), we cannot exclude that our results reflect some significant disparities in vaccination rates across France.

## Figures and Tables

**Figure 1 vaccines-10-01228-f001:**
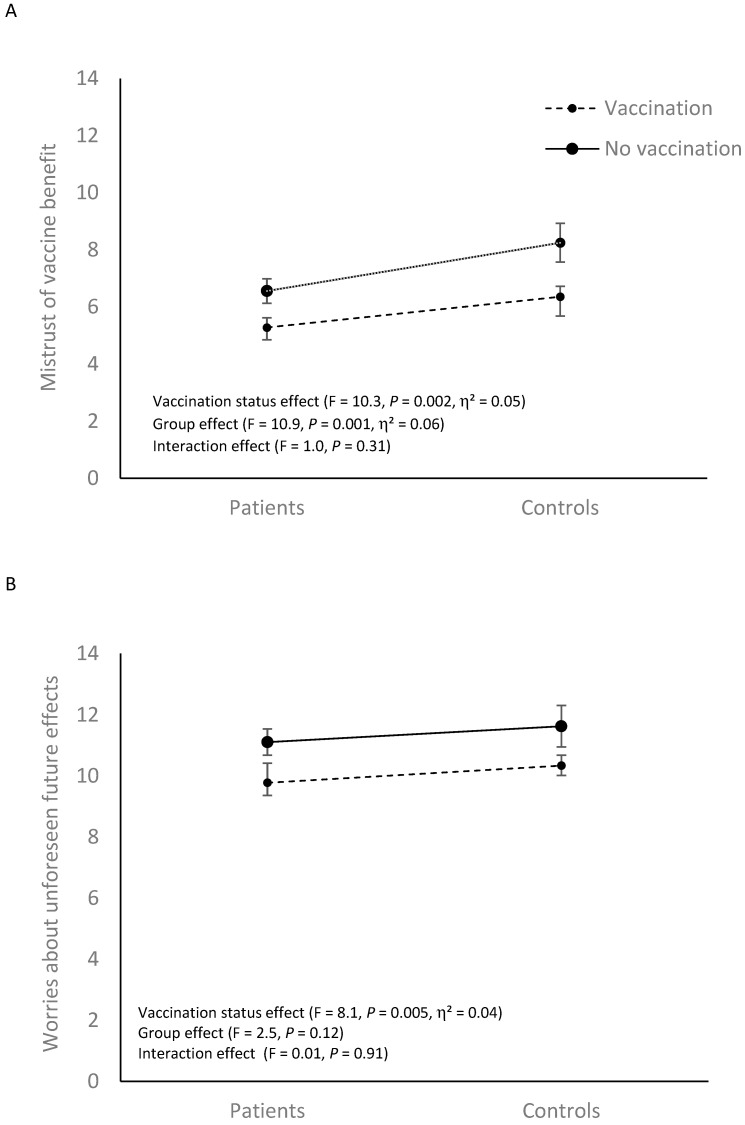
Vaccination Attitudes Examination Scale dimensions for patients and controls according to vaccinal status. Means (±standard error of the mean) are given. (**A**–**D**) correspond to the 4 dimensions of the Vaccination Attitudes Examination Scale.

**Table 1 vaccines-10-01228-t001:** Sociodemographic and clinical characteristics of the sample.

	Patients	Controls	Statistics	*p*-Value
*N* (%)/mean (SD)	*N* = 100	*N* = 72		
*Demographic variables*				
Age	40.6 (12.4)	32.7 (12.6)	t = −4.09	<0.001
Gender, female	37.50%	26%	χ^2^ = 2.09	0.15
Years of scholarship, mean	12.7 (2.7)	14.1 (2.2)	t = 3.76	<0.001
*Clinical and cognitive assessments*				
Patients under treatment	100%			
Duration of the disease	16.5 (11.3)			
PANSS total	63.9 (17.7)			
PANSS positive	13.7 (5.7)			
PANSS negative	16.8 (5.6)			
PANSS general psychopathology	33.2 (8.5)			
Green Paranoid Thoughts Scale	16.1 (8.6)	13.4 (6.9)	t = −2.16	0.02
Montreal Cognitive Assessment	23.9 (4.1)	27.1 (2.4)	t = 6.33	<0.001
*Vaccination status and health-related indicators*				
COVID-19 vaccination, yes	64%	77.80%	χ^2^ = 3.14	0.07
COVID-19 infection, yes	15%	23.60%	χ^2^ = 2.1	0.15
Underlying health condition, yes	22%	8.30%	χ^2^ = 5.74	0.01
Intend to take a COVID-19 vaccine ^1^	41.7%%	12.50%	χ^2^ = 4.28	0.03

Note. PANSS, Positive and Negative Syndrome Scale. ^1^ Among non-vaccinated participants (patients, *N* = 36; controls, *N* = 16).

## Data Availability

The anonymized dataset is available from the corresponding author on reasonable request.

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
