# Peer review of "Attitudes towards Vaccines, Intent to Vaccinate and the Relationship with COVID-19 Vaccination Rates in Individuals with Schizophrenia"

_vaccines, 2022, doi:10.3390/vaccines10081228_

Round 1

Reviewer 1 Report

I suggest to provide more information in the Methods: how patients and controlled were selected? randomly? From which setting? Institutionalised? If yes, what was the rate of vaccination among the target population? Where there any advocacy/communication/community engagement activities for the different groups? Was the exposure different?

I suggest to show it clearly in the right sections and the analyses. 

Author Response

Reviewer 1 : 

Comments and Suggestions for Authors

I suggest to provide more information in the Methods: how patients and controlled were selected? randomly? From which setting? Institutionalised? If yes, what was the rate of vaccination among the target population? Where there any advocacy/communication/community engagement activities for the different groups? Was the exposure different?

I suggest to show it clearly in the right sections and the analyses. 

Response: 

We thank the reviewer for this helpful comment.

We have added specifications on how patients were recruited on P.6 :

 «  

For the clinical participants, nursing staffs of the different sites identified all incoming outpatients with a diagnosis of schizophrenia.  In addition, the nursing staff regularly checked the electronic medical records and consultation boards for any potential clinical participants.  A total of 122 individuals with schizophrenia were screened for eligibility during the study period, amongst which 120 eligible patients were approached for an interview and a full description of the study. Mental capacity to participate to this study was based on the clinical assessment and a thorough review of the patients’ clinical of notes. All participants provided informed and written consent. Additional information was collected from the participants and they were then administered the measures, for a total duration of approximately one hour. Eighteen patients declined participation, and twelve did not complete the entire protocol. Therefore our final clinical sample included 100 individuals with a diagnosis of schizophrenia.

» 

We also added more details regarding the nonclinical group (P.6): « For the nonclinical group, participants were recruited from the general population using posters on the hospitals’ noticeboards and through word of mouth. » 

We thank Reviewer 1 for this comment that helped us improve the quality of the manuscript. We hope that the manuscript will be considered for publication in Vaccines in its new form.

Reviewer 2 Report

The authors have conducted a cross-sectional study of vaccination status and attitudes among individuals with schizophrenia and compared that to a random sample of individuals without schizophrenia. The study is quite straightforward. In addition to a final check on the prepositions used (a few mix-ups with as/than and in/for) I have a single suggestion to improve the article. Would the articles better describe how they recruited the 72 controls?

Author Response

Reviewer 2: 

Comments and Suggestions for Authors

The authors have conducted a cross-sectional study of vaccination status and attitudes among individuals with schizophrenia and compared that to a random sample of individuals without schizophrenia. The study is quite straightforward. In addition to a final check on the prepositions used (a few mix-ups with as/than and in/for) I have a single suggestion to improve the article. Would the articles better describe how they recruited the 72 controls?

Response : We thank Reviewer 2 for this positive comment and we have made the following changes. 

Following this suggestion, we added P.6: « For the nonclinical group, participants were recruited from the general population using posters on the hospitals’ noticeboards and through word of mouth. » 

We thank Reviewer 2 for the positive review and suggestions that helped us improve the quality of the manuscript. We hope that the manuscript will be considered for publication in Vaccines in its new form.

Reviewer 3 Report

Dear Authors,

I have reviewed with great interest the manuscript titled Attitudes towards vaccines, intent to vaccinate and the relationship with COVID-19 vaccination rates in individuals with schizophrenia. After evaluating the various points laid out in the article, I think that the authors have largely succeeded in producing a relevant, succinct and original piece of research. The article has several strengths: it is grounded in solid methodology and is competently assemble; the overall structure is coherent and rationally outlined throughout.

Most importantly, it does cover, as pointed ut by the authors themselves, an important and as yet rather underresearched correlation: mental health disorders and vaccination uptake/vaccine hesitancy. the table and figure are both effective at delineating key analytical aspects. 

A weakness that should in my view be addressed has to do with an inadequate degree of contextualization. The structural barriers that the authors mention as determining factors for the disparities in vaccination rates ought to be weighed against other factors that the authors disregarded or did not discuss in depth.

Studies have explored for instance the possibility of developing neuropsychiatric side-effects of vaccination and the potential interactions between vaccines and psychotropic medications. All such associations may play a role in determining lower uptake in mental patients, including schizophrenia sufferers. Furthermore, weighing demographic factors linked to mental health and vaccination acceptation may also be beneficial for the sake of thoroughness. Research shows that patients with severe mental illness have 2–3 times greater mortality due to COVID-19 than the general population, especially patients with schizophrenia spectrum disorders. Another element that the authors ought to take into account is the impact of social distancing, isolation and lockdowns on mental patients and how such dynamics may affect the willingness to get immunized and undermine the overall mental stability of such patients. Lastly, I would briefly discuss the likely effect of misinformation and anti-vaccination influences in schizophrenia patients. While none of those elements are the centerpiece of the article's foundational analysis, the authors have to take them into account when they discuss their own findings, since they are highly consequential contributing factors in shaping vaccination attitudes in mentally vulnerable patients.

the article is fairly well written overall, but I would recommend further proofreading from a native speaker of English in order to fix a few instances of clumsy grammar and questionable vocabulary choices.

I ultimately feel that the authors have made a commendable effort to shed light on a highly relevant area of mental research against the backdrop of the ongoing COVID-19 pandemic.

Sincerely,

Author Response

Reviewer 3: 

Comments and Suggestions for Authors

Dear Authors,

I have reviewed with great interest the manuscript titled Attitudes towards vaccines, intent to vaccinate and the relationship with COVID-19 vaccination rates in individuals with schizophrenia. After evaluating the various points laid out in the article, I think that the authors have largely succeeded in producing a relevant, succinct and original piece of research. The article has several strengths: it is grounded in solid methodology and is competently assemble; the overall structure is coherent and rationally outlined throughout.

Most importantly, it does cover, as pointed ut by the authors themselves, an important and as yet rather underresearched correlation: mental health disorders and vaccination uptake/vaccine hesitancy. the table and figure are both effective at delineating key analytical aspects. 

A weakness that should in my view be addressed has to do with an inadequate degree of contextualization. The structural barriers that the authors mention as determining factors for the disparities in vaccination rates ought to be weighed against other factors that the authors disregarded or did not discuss in depth.

Studies have explored for instance the possibility of developing neuropsychiatric side-effects of vaccination and the potential interactions between vaccines and psychotropic medications. All such associations may play a role in determining lower uptake in mental patients, including schizophrenia sufferers. Furthermore, weighing demographic factors linked to mental health and vaccination acceptation may also be beneficial for the sake of thoroughness. Research shows that patients with severe mental illness have 2–3 times greater mortality due to COVID-19 than the general population, especially patients with schizophrenia spectrum disorders. 

Response. 

As suggested by reviewer 3, we have added the following in the discussion section P. 9 :

« Finally we did not take into account other non-structural determining factors that could have contributed to the differences in vaccination rates in our two populations. For example, several studies have explored the risk of developing neuropsychiatric side-effects of vaccination and the potential interactions between vaccines and psychotropic medications [43]. Such side-effects may have contributed to lower vaccine uptake in our clinical sample and may represent an additional limitation to our study. »

Another element that the authors ought to take into account is the impact of social distancing, isolation and lockdowns on mental patients and how such dynamics may affect the willingness to get immunized and undermine the overall mental stability of such patients. 

Response: We discussed this point in the discussion P8-9:

« The association between negative symptoms and vaccination hesitancy appears however difficult to explain and remains elusive. Nevertheless, the desires for restrictions to ease and life to return to normal (e.g. go to the cinema, travel) has been found as the most common reasons for positive attitudes towards COVID-19 vaccine [37, 38]. It is likely that the desire to perform such activities do not constitute a strong enough factor in reducing vaccine hesitancy in people with schizophrenia with high levels of negative symptoms.  For example, studies have recurrently shown that negative symptoms, and specifically apathy-anhedonia, lead to a reduced capacity in experiencing and anticipating pleasure from activities [39]).   A reduction in interpersonal interactions and goal-directed behaviors have also been noted in this population.  This could also explain the association found between negative symptoms and vaccination hesitancy. » 

Lastly, I would briefly discuss the likely effect of misinformation and anti-vaccination influences in schizophrenia patients. While none of those elements are the centerpiece of the article's foundational analysis, the authors have to take them into account when they discuss their own findings, since they are highly consequential contributing factors in shaping vaccination attitudes in mentally vulnerable patients.

Response: 

We have discussed briefly this point in the introduction P. 2 : 

« Recently and within the context of the COVID-19 pandemic, two large-scale studies have shown that both conspiracy beliefs and persecutory ideas, two psychological features that are highly present in individuals with schizophrenia, were positively associated with COVID-19 vaccine hesitancy and resistance in the general population [15,16]. »

And in the discussion P. 8

« Fourth, correlational analyses found negative relationships between positive attitudes toward vaccination and negative symptoms and paranoid ideations, measured using the PANSS and the GPTS, respectively. The symptomatology of individuals with schizophrenia should thus be considered when assessing confidence and intent towards vaccination. The association between paranoid ideation and general negative attitudes towards vaccination in schizophrenia is in accordance with previous studies conducted in the general population whereby similar associations between conspiracy beliefs, persecutory beliefs and higher resistance to a COVID-19 vaccine was found [15,16]. However, the fact that general attitudes towards vaccination were similar between patients and controls suggests that individuals with schizophrenia are not additionally impacted by misinformation and anti-vaccination influences compared to the general population. »

Following the reviewer’s remark and in accordance with the reviewer 5 we also added the following on

P.8:

« Our results also indicate that non-vaccinated participants (in both groups), had lower levels of trust in institutions compared to the vaccinated participants. This result is in accordance with recent research that emphasized the role of trust in science as a determinant factor in both vaccine hesitancy and intention to be vaccinated [40, 41, 42]. Importantly, this result shows similar associations in individuals with schizophrenia. » 

the article is fairly well written overall, but I would recommend further proofreading from a native speaker of English in order to fix a few instances of clumsy grammar and questionable vocabulary choices.

Response: 

Thank you for this comment.  We have proofread and revised our manuscript accordingly.

I ultimately feel that the authors have made a commendable effort to shed light on a highly relevant area of mental research against the backdrop of the ongoing COVID-19 pandemic.

Sincerely,

Response: We thank Reviewer 3 for the comments that have helped us improve the quality of the manuscript. We hope that the manuscript will be considered for publication in Vaccines in its new form.

Reviewer 4 Report

It is remarkable that the investigators have been able to produce such high-quality research in what is a population that is difficult to study, a population that at the same time is at increased risk of SARS-CoV-2 infection and with a risk of increased levels of vaccine hesitancy and refusal. Of course it is difficult to tease out whether different degrees of schizophrenia and different medications confound the results. Nevertheless, the rigorous statistical approach reassures the reader that the results are valid.

It is helpful to know that structural changes could result in improved vaccination coverage, and that these patients are highly willing to be vaccinated. One suspects that the results could be applied to disease areas beyond SARS-CoV-2.

Author Response

Reviewer 4:

Comments and Suggestions for Authors

It is remarkable that the investigators have been able to produce such high-quality research in what is a population that is difficult to study, a population that at the same time is at increased risk of SARS-CoV-2 infection and with a risk of increased levels of vaccine hesitancy and refusal. Of course it is difficult to tease out whether different degrees of schizophrenia and different medications confound the results. Nevertheless, the rigorous statistical approach reassures the reader that the results are valid.

It is helpful to know that structural changes could result in improved vaccination coverage, and that these patients are highly willing to be vaccinated. One suspects that the results could be applied to disease areas beyond SARS-CoV-2.

Response.

We thank Reviewer 4 for this positive feedback. We hope that the manuscript will be considered for publication in Vaccines in its new form.

Reviewer 5 Report

Vaccines

Attitudes towards vaccines, intent to vaccinate and the

relationship with COVID-19 vaccination rates in individuals with schizophrenia

The present work investigates the role of potential attitudinal barriers by comparing schizophrenia patients with participants from the general population regarding their general attitude towards the COVID-19 vaccine, in particular the willingness to take it and level of acceptance. The research was conducted cross-sectionally between April 2021 and 30 October 2021 and compared patients with schizophrenia (N=100) and controls (N=72). The main result that emerges is that vaccine hesitancy does not appear to be a major barrier to COVID-19 vaccine uptake among patients with schizophrenia. Furthermore, the inclusion of the difference between structural and attitudinal barriers to vaccination in this study seems to demonstrate that the disparity in COVID-19 vaccination rates in schizophrenia does not appear to be related to attitudinal but rather structural barriers.

I find this topic extremely interesting and believe the authors have addressed it competently. The introduction provides a clear summary of the state of the art in the field and a coherent framework for the authors' hypotheses; the methods section is rigorously described, the data analyses are clear, and I appreciate the authors' efforts to provide transparent data.

Nevertheless, I have some concerns about the interpretation of the results. I believe that the authors need to revise the Discussion section for the article to be accepted for publication. In the following paragraphs I address this issue, along with some minor comments to improve the readability and understanding of the article.

Introduction:

·       The introduction of the article follows a clear structure. The topic of vaccination and the relevant studies in the literature on schizophrenic patients are described in a comprehensive manner. The distinction between structural and attitudinal barriers in vaccination hesitation is also clear. However, a more accurate and detailed description of the study cited on line 83-84 is needed. Here the authors quote 'various mental disorders' which need to be better described, for the reader to understand the comparison with other mental disorders.

 Material and Methods:

·       I would recommend performing an effect size analysis to determine whether the power was sufficient to detect between-group differences

·       Line 132: please specify the meaning of ‘fully vaccinated’? Two or three doses?

·       Attitude towards vaccine questionnaire. Please provide examples of the items for the four subscales of the Vaccination Attitudes Examination Scale.

·       Please provide the Cronbach αs obtained in the present study for the Vaccination Attitudes Examination Scale and the Trust in institutions questionnaire. This is important to verify that internal consistency was adequate.

Statistical analyses

·       The skewness and kurtosis values adopted in the present study (>3 and >20 respectively) are very high: according to many authors, values for asymmetry and kurtosis between -2 and +2 are considered acceptable in order to prove normal univariate distribution (George & Mallery, 2010). Hair et al. (2010) and Bryne (2010) argued that data is considered to be normal if skewness is between 2 to +2 and kurtosis is between 7 to +7. I think that the authors should explicitly report the current values of skewness and kurtosis for continuous variable and use lower critical thresholds.

·       Winsorizing method: What was the definition of outlier? This is not clear.

Results:

·       Since the two groups differed significantly in terms of age and education, these two variables should be included as covariates in all the ANOVA analyses in which the two groups were directly compared.

Discussion:

In general, I found the discussion easy to follow. I suggest referring to the results section more clearly to help the reader better understand the interpretation of the data. However, I feel that the section of the discussion on the main objective of the study should be justified with an interpretation of the results based on the literature. For example:

·       I would provide a stronger explanation as to why there is no significant difference between patients and controls regarding intention to vaccinate.

·       Another element: I would suggest the authors to consider recent works that emphasised the role of trust in science in determining both vaccine hesitancy and intention to be vaccinated (e.g., Kerr et al., 2021; Santirocchi et al., 2022; Viswanath et al., 2021). I think that replicating this association in schizophrenic patients is important and, even if the finding is correlational in nature, it should be discussed considering the recent studies available in the literature.

Author Response

Reviewer 5: 

Comments and Suggestions for Authors

The present work investigates the role of potential attitudinal barriers by comparing schizophrenia patients with participants from the general population regarding their general attitude towards the COVID-19 vaccine, in particular the willingness to take it and level of acceptance. The research was conducted cross-sectionally between April 2021 and 30 October 2021 and compared patients with schizophrenia (N=100) and controls (N=72). The main result that emerges is that vaccine hesitancy does not appear to be a major barrier to COVID-19 vaccine uptake among patients with schizophrenia. Furthermore, the inclusion of the difference between structural and attitudinal barriers to vaccination in this study seems to demonstrate that the disparity in COVID-19 vaccination rates in schizophrenia does not appear to be related to attitudinal but rather structural barriers.

I find this topic extremely interesting and believe the authors have addressed it competently. The introduction provides a clear summary of the state of the art in the field and a coherent framework for the authors' hypotheses; the methods section is rigorously described, the data analyses are clear, and I appreciate the authors' efforts to provide transparent data.

Nevertheless, I have some concerns about the interpretation of the results. I believe that the authors need to revise the Discussion section for the article to be accepted for publication. In the following paragraphs I address this issue, along with some minor comments to improve the readability and understanding of the article.

Introduction:

·       The introduction of the article follows a clear structure. The topic of vaccination and the relevant studies in the literature on schizophrenic patients are described in a comprehensive manner. The distinction between structural and attitudinal barriers in vaccination hesitation is also clear. However, a more accurate and detailed description of the study cited on line 83-84 is needed. Here the authors quote 'various mental disorders' which need to be better described, for the reader to understand the comparison with other mental disorders.

Response: We thank the reviewer for this comment. 

Following the suggestion, we added the following sentence : « However, the majority of their sample was composed of people diagnosed with anxiety disorders (including generalized anxiety disorder and panic disorder) and depression (major depressive disorder) with only a small sample (N=16) of individuals diagnosed with psychotic disorders [19, 20]. »

 Material and Methods:

·       I would recommend performing an effect size analysis to determine whether the power was sufficient to detect between-group differences.

Response: Two power analyses were conducted using G*Power version 3.1.9.7 (Faul et al., 2007) for sample size estimation (see below for the output of theses analyses). The effect sizes chosen were either 0.25 or 0.40, considered medium and large effect sizes using Cohen's (1988) criteria. With a significance criterion of α = 0.05 and power = 0.80, the minimum sample size needed with these effect sizes is N = 211 (medium effect) and N = 86 (large effect) for the ANCOVA (with two covariates) recommended by the Reviewer 5. Our sample was N = 172, which appears to be sufficient.

·       Line 132: please specify the meaning of ‘fully vaccinated’? Two or three doses?

Response: In our text, full vaccination signified the administration of all three doses of vaccine. The manuscript has been modified accordingly:

Vaccination status and health-related indicators. COVID-19 vaccination status was recorded for all participants as either no vaccination or first dose of vaccine/fully vaccinated (i.e., three doses). […]

·       Attitude towards vaccine questionnaire. Please provide examples of the items for the four subscales of the Vaccination Attitudes Examination Scale.

Response: Examples of the items were provided in the Assessment section. The manuscript has been modified accordingly.

[…] Responses on the VAX are rated on a six-point scale ranging from 1 (strongly agree) to 6 (strongly disagree). Four subscale-scores were calculated: (1) mistrust of vaccine benefit (e.g. “I can count on vaccines to remove serious infectious diseases”, reverse item), (2) worries about unforeseen future effects (e.g., “I concerned about the unknown long-term effects of effects of vaccines”), (3) concerns about commercial profiteering (e.g., “The authorities encourage vaccination for financial reasons, not for the health of the people”), and (4) preference for natural immunity (e.g., “Natural immunity lasts longer than than that acquired by vaccination”). […]

·       Please provide the Cronbach αs obtained in the present study for the Vaccination Attitudes Examination Scale and the Trust in institutions questionnaire. This is important to verify that internal consistency was adequate.

Response: The Cronbach α’s were computed for this present study. We also computed the corrected item-to-total correlation. The Assessment and Preliminary analyses sections have been modified accordingly.

Assessment section: 

[…] The internal consistency of the French version of the VAX was examined with the corrected item-to-total correlation and Cronbach’s alpha (good internal consistency was considered 0.7 > α < 0.9). […]

Preliminary analyses sections:

[…] Overall, the internal consistency of the French version of the VAX was good. Except for the worries about unforeseen future effects (α = 0.55) subscale, all Cronbach’s alpha for the other subscales of the VAX were considered good: mistrust of vaccine benefit (α = 0.86), concerns about commercial profiteering (α = 0.74), and preference for natural immunity (α = 0.75) subscales. The item-total correlations for the 14 items of the VAX ranged from 0.35 to 0.66, with a mean of 0.54.

Statistical analyses

·       The skewness and kurtosis values adopted in the present study (>3 and >20 respectively) are very high: according to many authors, values for asymmetry and kurtosis between -2 and +2 are considered acceptable in order to prove normal univariate distribution (George & Mallery, 2010). Hair et al. (2010) and Bryne (2010) argued that data is considered to be normal if skewness is between ‐2 to +2 and kurtosis is between ‐7 to +7. I think that the authors should explicitly report the current values of skewness and kurtosis for continuous variable and use lower critical thresholds.

Response: There is indeed no consensus on the threshold values of the skewness and kurtosis indices. There is significant variability among authors regarding these values. In our publications, we usually choose the reference of Weston recommended by Andy Field who is a reference author in statistics. In our study and as documented in the table just below, the skewness and kurtosis values of our variables of interest varied between -0.92 and 3.24. These values are acceptable in relation to the references cited by Reviewer 5.

Age

Education

VAX

Gpts

PANSS

MOCA

Benefit

Worries

Profit

Immuno.

Total

Positive

Negative

Psychopatho.

Skewness

0.32

0.01

0.90

-0.35

0.68

0.20

1.09

1.18

1.38

0.92

0.92

1.15

Kurtosis

0.74

-0.92

0.39

-0.20

-0.06

0.09

0.09

3.24

2.25

1.70

1.70

1.11

We propose to keep the reference originally cited and to briefly describe these values in the Preliminary analyses section. The manuscript has been modified accordingly:

All data from questionnaires had satisfactory skewness (0.01 to 1.38) and kurtosis (-0.92 to 3.24) values, suggesting a normal distribution.

·       Winsorizing method: What was the definition of outlier? This is not clear.

Response: We have provided the definition of what we consider to be an outlier value. The Statistical analysis section has been modified accordingly.

The winsorizing method was used to process any outlier scores on questionnaires. An outlier score was considered when its z-score had an absolute value of |3.29| which is 0.01% of the distribution. This method consists of replacing outlier scores by assigning a value on a unit smaller or larger than the next most extreme (non-outlier) score in the distribution [28].

Results:

·       Since the two groups differed significantly in terms of age and education, these two variables should be included as covariates in all the ANOVA analyses in which the two groups were directly compared.

Response: We are surprised that the values of the analyses of variance that we had embedded directly in the figures, did not appear in the manuscript in the following way:

ORIGINAL VERSION - ANOVA

Figure 1. Vaccination Attitudes Examination Scale dimensions for patients and controls according to vaccinal status. Means (± standard error of the mean) are given.

A

B

C

D

As requested by Reviewer 5, we redid the analyses of variance considering age and education as covariates. The pattern of results are similar compared to the original analysis of variance. 

REVISED VERSION - ANCOVA

Figure 1. Vaccination Attitudes Examination Scale dimensions for patients and controls according to vaccinal status. Means (± standard error of the mean) are given.

A

B

C

D

Discussion:

In general, I found the discussion easy to follow. I suggest referring to the results section more clearly to help the reader better understand the interpretation of the data. However, I feel that the section of the discussion on the main objective of the study should be justified with an interpretation of the results based on the literature. For example:

·       I would provide a stronger explanation as to why there is no significant difference between patients and controls regarding intention to vaccinate.

    Another element: I would suggest the authors to consider recent works that emphasised the role of trust in science in determining both vaccine hesitancy and intention to be vaccinated (e.g., Kerr et al., 2021; Santirocchi et al., 2022; Viswanath et al., 2021). I think that replicating this association in schizophrenic patients is important and, even if the finding is correlational in nature, it should be discussed considering the recent studies available in the literature.

Response: 

Thank you for this comment, we agree with the reviewer’s remark. We added the following on P. 8:

« One can also hypothesize that the high rate of positive attitudes towards vaccination may partly be explained by the prioritization for vaccination of patients with mental disorders following the publication of high mortality rates of these populations in France [2]. »

We also added the following on P. 8

« Finally, we also found that individuals with schizophrenia had significantly higher level of confidence in the institutions in comparison to the controls.  Our results also indicate that non-vaccinated participants (in both groups), had lower levels of trust in institutions compared to the vaccinated participants. This result is in accordance with recent research that emphasized the role of trust in science as a determinant factor in both vaccine hesitancy and intention to be vaccinated [40, 41, 42]. Importantly, this result shows similar associations in individuals with schizophrenia. »

We thank Reviewer 5 for the positive review that have helped us improve the quality of the manuscript. We hope that the improved version of the manuscript will be considered for publication in Vaccines.

Round 2

Reviewer 1 Report

No futher comments

Reviewer 3 Report

Dear Authors,

After reviewing the new version of your article, I do believe that you deserve credit for having improved the substance, breadth, and presentation of your writing. The overall scope of the article has been broadened enough for it to be comprehensive and well-rounded enough.

By virtue of its significance, thoroughness and originality, I am going to greenlight this article for publication.

Sincerely,